# Cardiac Complications of COVID-19 in Low-Risk Patients

**DOI:** 10.3390/v14061322

**Published:** 2022-06-17

**Authors:** Akash Srinivasan, Felyx Wong, Liam S. Couch, Brian X. Wang

**Affiliations:** 1Department of Medicine, Faculty of Medicine, Imperial College London, London SW7 2AD, UK; akash.srinivasan17@imperial.ac.uk (A.S.); felyx.wong17@imperial.ac.uk (F.W.); 2King’s College London BHF Centre, The Rayne Institute, St Thomas’ Hospital, London SE1 7EH, UK; liamscouch@gmail.com; 3Department of Metabolism, Digestion and Reproduction, Faculty of Medicine, Imperial College London, London SW7 2AZ, UK

**Keywords:** COVID-19, post-COVID syndrome, PIMS, SARS-CoV-2, cardiovascular disease

## Abstract

The coronavirus disease 2019 (COVID-19) pandemic has resulted in over 6 million deaths and significant morbidity across the globe. Alongside common respiratory symptoms, COVID-19 is associated with a variety of cardiovascular complications in the acute and post-acute phases of infection. The suggested pathophysiological mechanisms that underlie these complications include direct viral infection of the myocardium via the angiotensin-converting enzyme 2 (ACE2) protein and a cytokine release syndrome that results in indirect inflammatory damage to the heart. Patients with pre-existing cardiovascular disease and co-morbidities are generally more susceptible to the cardiac manifestations of COVID-19. However, studies have identified a variety of complications in low-risk individuals, including young adults and children. Myocarditis and paediatric inflammatory multisystem syndrome temporally associated with COVID-19 (PIMS) are among the adverse events reported in the acute phase of infection. Furthermore, patients have reported cardiac symptoms persisting beyond the acute phase in post-COVID syndrome. This review summarises the acute and chronic cardiac consequences of COVID-19 in low-risk patients, explores the pathophysiology behind them, and discusses new predictive factors for poor outcomes.

## 1. Introduction

Since its discovery in Wuhan in 2019 [1], the severe acute respiratory syndrome coronavirus 2 (SARS-CoV-2) has spread globally and created the worst pandemic since 1918 [2]. Infection with SARS-CoV-2 causes coronavirus disease 2019 (COVID-19), which has contributed to over 6 million deaths as of March 2022 [3] and has significant associated morbidity [4]. Acute kidney injury, acute respiratory distress syndrome, thrombosis, and cardiac injury are among the most frequent complications of the disease [4,5,6,7].

Potential long-term cardiac complications following COVID-19 in post-COVID syndrome include ischaemic heart disease, arrhythmias, and myocarditis [8]. Patients with pre-existing cardiovascular disease are at higher risk [9,10], but there are reports of younger patients with no known comorbidities presenting with concerning cardiac manifestations [11,12]. These findings support the evidence an increasing number of people experiencing a post-COVID syndrome, often termed “long COVID” [13], which is generating concern throughout the medical community.

Angiotensin-converting enzyme 2 (ACE2) plays a crucial role in the pathogenesis of COVID-19. ACE2 is highly expressed in the lungs, and binding of the virus to ACE2 plays a pivotal role in the development of respiratory insufficiency reported in many COVID-19 patients [14]. Death is a common consequence of pulmonary bacterial superinfection or extensive alveolar damage in post-COVID syndrome [15,16]. ACE2 is also expressed in the myocardium [17], and this may play an important role in the pathophysiology of acute cardiac complications secondary to COVID-19 [16,18].

This review summarises the acute and long-term cardiac manifestations of COVID-19 in patients considered to be less vulnerable to the disease, summarises the pathophysiology behind these complications, and explores the predictive factors for these unexpected cardiac consequences.

## 2. Pathophysiology of Cardiac Complications

The mechanisms underlying the cardiac complications of COVID-19 are becoming increasingly understood. In a similar manner to SARS-CoV and other coronaviruses, SARS-CoV-2 binds to ACE2 receptors, leading to cellular infection via endocytosis (Figure 1) [14]. ACE2 is primarily expressed in type 2 pneumocytes of the lung, hence the predilection of COVID-19 for presenting with signs of respiratory infection [19]. However, the potential for direct cardiac infection, direct cardiotoxicity, and myocarditis was illustrated following the discovery of ACE2 in the heart, with ACE2 being present in over 7.5% of the cells in the myocardium [19]. SARS-CoV-2 also demonstrates a greater affinity for ACE2 than previous coronaviruses, which may further facilitate cardiac infection [20], with viral particles detected in the heart after autopsy [21]. SARS-CoV-2 may induce apoptosis of endothelial cells, causing microvascular dysfunction, exposing the pro-coagulant basement membrane and promoting thrombosis [22]. This may provide a direct mechanism for COVID-related supply–demand mismatch and hypoxaemic damage or acute coronary syndromes, but remains to be fully explored.

Direct cardiac infection may also lead to post-COVID syndrome, as it can lead to cell death, lingering tissue damage, and impaired cardiac function. Research has shown that infection by the virus is associated with downregulation of ACE2 in endothelial cells [23], a finding that was previously linked to TGF-β signalling via the Smad pathway and renal fibrosis [24]. Conversely, upregulation of ACE2 was associated with reduced TGF-β and Smad2/3 expression, and attenuated myocardial fibrosis in rodents [25]. Taken together, this suggests ACE2 downregulation in humans may result in scarring of the myocardium and contribute to the common symptoms experienced by patients with post-COVID syndrome, such as fatigue and dyspnoea [13]. Additionally, myocardial fibrosis is a significant risk factor for arrhythmias, due to the non-excitable extracellular matrix interfering with electrical conduction, and this may explain reports of palpitations associated with post-COVID syndrome [26].

Conversely, many of the reported acute cardiac manifestations of COVID-19 can also be explained by indirect inflammatory mechanisms. Severe COVID-19 infection is characterised by the development of cytokine release syndrome and excessive inflammation [27], which are associated with poor patient outcomes, including mechanical ventilation and death [28]. Cytokine release syndrome manifests as acute respiratory distress syndrome (ARDS) in the lungs and drives inflammation in the tissue and vasculature of the heart [29]. Vascular inflammation promotes destabilisation of pre-existing atheromatous plaques and subsequent myocardial infarction [29], or may lead to microvascular dysfunction, supply–demand mismatch, and further hypoxaemic damage. Tissue inflammation also provides an alternative explanation for the reported cases of myocarditis [30,31]. Furthermore, the excessive release of cytokines was linked to arrhythmia generation, which may occur due to abnormal regulation of intracellular calcium ions [32]. The reports of cardiac injury in young patients without cardiovascular comorbidities, as previously discussed, suggests that the severity of cytokine release syndrome alone in COVID-19 may be sufficient to cause extensive damage to an otherwise healthy heart. Persistent inflammation may also be responsible for a variety of issues in post-COVID syndrome. Potential causes of a prolonged immune response include the persistence of SARS-CoV-2 in tissues where ACE2 is highly expressed, such as the heart [33], or the integration of the COVID-19 RNA into the human genome [34]. Nevertheless, further research is required to gain a clearer understanding of the mechanisms underlying acute and chronic cardiac complications with COVID-19 and its wider effects on the cardiovascular system.

## 3. Acute Cardiac Complications

### 3.1. Low-Risk Adults

Patients at low-risk for cardiac complications from COVID-19 are generally younger, with a meta-analysis estimating a two-fold higher incidence of cardiac injury in hospitalised patients over the age of 60 years [35]. Immediately following COVID-19 infection, the common cardiac manifestations in adults include acute coronary syndrome and arrhythmia [36]. These complications are predominantly associated with known cardiovascular risk factors, including hypertension, diabetes mellitus, and pre-existing coronary heart disease [10,17]. However, different acute cardiac complications were observed in low-risk adults without comorbidities, such as acute myopericarditis [37].

Recently, 67 patients with COVID-19 and acute (myo)pericarditis were analysed in a case-control study [38], and the prevalence of common comorbidities, including hypertension, dyslipidaemia, diabetes, and obesity, was lower in patients with acute (myo)pericarditis than COVID-19 patients without acute (myo)pericarditis [38]. Thus, standard cardiovascular risk factors may be poor predictors of developing myopericarditis with COVID-19, although further research with larger cohorts is required to validate this observation. Furthermore, a recent systematic review of 41 case reports and case series calculated that the median age of reported COVID-19 patients with myocarditis was 43 years [39]. This number may be influenced by publication bias: COVID-19 cases involving younger patients with unexpectedly severe outcomes are more likely to be reported in the literature. Nevertheless, a large-scale study estimated that for individuals younger than 40 years, there are 10 additional cases of myocarditis per 1 million people with a positive COVID-19 test [40]. Thus, COVID-associated myocarditis appears to be a genuine threat to younger adults without comorbidities. 

Increased incidence of takotsubo syndrome has also been reported during COVID-19 infection [41], with cytokine storm, increased inflammation, endothelial dysfunction, and increased sympathetic responses during acute COVID-19 infection proposed as underlying this observation in the recent expert consensus on the pathophysiology of takotsubo syndrome [42,43]. This may also result from increased acute and chronic stressors present throughout the pandemic, with pre-existing neuropsychiatric stress thought to increase the risk of future development of takotsubo syndrome [43,44].

Cardiac biomarkers are typically deranged during the acute phase of COVID-19 in hospitalised patients, with troponin and B-type natriuretic peptide (BNP) levels elevation particularly associated with mortality [45,46]. N-terminal prohormone of BNP is also important in the diagnosis of acute (myo)pericarditis associated with COVID-19, as the patients do not always experience significant symptoms [38]. Transthoracic echocardiography (TTE) is the imaging modality of choice during acute infection, primarily for suspected heart failure [47], and may reveal ventricular dilatation and dysfunction [48]. Notably, these abnormalities are less common but still evident in patients without pre-existing cardiac disease [47]. The use of TTE can aid the diagnosis of acute coronary syndrome, takotsubo syndrome, and myocarditis [47], and abnormal findings are associated with a worse prognosis [49].

### 3.2. Children

Mortality data from England between March 2020 and February 2021 showed that individuals under the age of 18 years with a positive COVID-19 test had a survival rate of 99.995% [50]. This was considerably higher than the overall survival rate during the same time period [51]. The most common symptoms experienced by children include fever, cough, nausea, and diarrhoea, whilst a significant proportion usually remain asymptomatic [52]. However, despite the early indications that children were less susceptible to the virus [53], new concerns were raised about potential cardiac involvement when clusters of children developed a Kawasaki-like disease during the first wave of the pandemic [54]. This phenomenon has been explored further and is known as paediatric inflammatory multisystem syndrome temporally associated with COVID-19 (PIMS) [55,56].

One of the earliest case series examined a 10-day period during April 2020, and identified a cluster of eight children in England who developed hyperinflammatory shock and symptoms suggestive of an atypical Kawasaki disease [57]. One child developed arrhythmia and passed away, with the post-mortem detecting SARS-CoV-2 infection [57]. Thereafter, another patient developed a coronary artery aneurysm, a known complication of Kawasaki disease [57]. Another cohort in Italy also detected a markedly increased incidence of Kawasaki-like disease at the beginning of the pandemic, with the affected patients having a mean age of 7.5 years and more frequent cardiac complications versus classic Kawasaki disease [54]. Subsequently, PIMS was defined and reports of this complication emerged in many countries, including South Africa [58], the USA [59], and other European nations [60,61].

Key differences emerged between the symptoms of PIMS versus Kawasaki disease, as highlighted in Table 1. Gastrointestinal symptoms such as abdominal pain, vomiting, and diarrhoea were unusually common [62], and cardiac involvement was more frequent in PIMS [54]. The patients with PIMS were also older than expected for Kawasaki disease, with an average age of 7 to 8 years for PIMS versus 3 years old for Kawasaki disease [54,57,62].

Analysis of 58 hospitalised children with PIMS showed that 14% developed dilatation or aneurysms of the coronary arteries [11]. Compared with Kawasaki disease, this occurrence was in older children who had significantly raised cardiac markers and greater levels of inflammation [11]. However, this study failed to identify any predictive factors for the development of coronary artery aneurysms [11]. Further cardiac complications were identified in PIMS, including valvular regurgitation and left ventricular dysfunction [55]. Despite the severity of the initial illness, it is unclear whether there is significant long-term morbidity in patients with PIMS, although a recent follow-up study found no echocardiographic abnormalities in 96% of patients up to 6 months afterward [56].

Inflammatory markers including C-reactive protein, D-dimer, fibrinogen, and interleukin-6 are usually markedly raised in PIMS, and the elevation tends to be proportional to severity [11,63]. Troponin and BNP may also be raised in severe disease, indicating cardiac involvement [55]. Further cardiac assessment involves performing electrocardiography, as patients can develop first-degree atrioventricular block [64], and echocardiography to assess for coronary artery aneurysms and ventricular dysfunction [65]. Additional echocardiography findings may include pericardial effusion and mitral regurgitation [65].

The high rates of COVID-19 transmission between predominantly unvaccinated children justifies the need to identify those at greater risk of developing severe complications, such as PIMS. Studies have indicated that PIMS disproportionately affects children of Afro-Caribbean or Hispanic descent [11,62,66]. Similarly, a U.K. PIMS national surveillance programme found that Black and Asian patients were grossly overrepresented in the PIMS cohort relative to the overall population, although possible confounding factors were present, such as the increased likelihood of parents being key workers [67]. Patients of these ethnicities also appear to be at greater risk of severe disease and higher paediatric intensive care admission after COVID-19 infection [68]. Interestingly, children from Asia outside of the Indian sub-continent appear to be less likely to develop PIMS, which is in contrast with the epidemiology associated with Kawasaki disease [67,69]. Genetic risk factors are also likely to be significant, with whole-exome sequencing identifying PIMS-associated mutations in the DOCK8 intracellular signalling proteins, cytochrome b-245 subunits, and X-linked inhibitors of apoptosis [70]. However, it is still unclear as to how these mutations increase the risk of PIMS. Hence, further research is necessary before targeted prophylactics and therapies can be designed.

## 4. Post-Acute Cardiac Sequelae

Multiple studies have highlighted persisting COVID-19 symptoms in post-COVID syndrome and the development of atypical symptoms after the acute period [71,72], which can occur in patients who initially had a mild or asymptomatic infection [73]. In the U.K., the National Institute for Health and Care Excellence (NICE) describes “long COVID” as the presence of new or ongoing symptoms 4 weeks or more after the onset of acute COVID-19 [74]. The most common symptoms of long COVID include fatigue, malaise, and cognitive dysfunction [13], but reports of palpitations [75] and chest pain [76] have generated concern surrounding the possible cardiac consequences of long COVID.

In a large study comparing the cardiovascular outcomes 30 days post-infection in 153,760 patients with COVID-19, patients experienced a 72% higher risk of heart failure, 63% higher risk of myocardial infarction, and higher rates of dysrhythmias and inflammatory heart disease versus contemporary and historical controls [8]. Notably, these increased risks were present across all age groups and in patients who had no prior cardiovascular disease [8]. A prospective study of 201 patients also identified cardiac dysfunction as a potential feature of long COVID, with myocarditis or systolic dysfunction identified in 26% of participants at 4 months [77]. These patients were generally considered low-risk, with a mean age of 45 years and only a 5% presence of pre-existing heart disease [77]. The incidence of cardiac complications observed in this study is strikingly high and highlights the long-term threat of COVID-19 to otherwise healthy individuals.

Analysis of data collected predominantly in the United Kingdom by the COVID Symptom Study application highlighted chest pain as a common symptom of long COVID, with a median duration of over 40 days [78]. Additional studies have shown chest pain to be present in up to 20% of cases after 60 days [79,80]. Notably, a follow-up study of 1733 discharged COVID patients found that 5% of participants reported chest pain at 6 months following acute infection [81]. In one case report, exertional chest pain was reported at 1 month after infection in a woman with no previous history of cardiovascular disease [76]. Although the underlying pathophysiology of chest pain in long COVID is unclear, evidence of coronary microvascular dysfunction was identified by adenosine stress cardiovascular magnetic resonance imaging (CMR), and the authors concluded that prolonged chest pain associated with long COVID may be a consequence of coronary microvascular ischaemia [76]. Another study utilising CMR found signs of persistent myocardial inflammation in 60% of recovered participants, suggesting an alternative mechanism for chest pain in long COVID [12].

Palpitations have also been reported by 20% of patients at 71 days [12] and 9% at 6 months after COVID-19 infection [81]. A small proportion of participants in the COVID Symptom Study application analysis also reported palpitations and tachycardia at 4 weeks [78]. Of interest, a 26 year old female patient with no co-morbidities aside from asthma reported palpitations persisting for over 5 months after testing positive for COVID-19, which resulted from developing autonomic dysfunction and postural tachycardia syndrome [75]. There are further reports of autonomic dysfunction amongst recovered patients after the SARS epidemic [82], suggesting that this may be a common characteristic of SARS-CoV-2 that precipitates this phenomenon.

The diagnostic approach to post-COVID syndrome depends on the type of symptoms present; regular 12-lead electrocardiography may be indicated for chest pain, whereas extended Holter monitoring aids the assessment of postural tachycardia syndrome [83]. TTE may be performed in suspected myocardial injury, but it has limited diagnostic utility for myocarditis in post-COVID syndrome [84]. Finally, studies have highlighted CMR findings associated with post-acute cardiac manifestations, including raised naïve T1 and T2, and late gadolinium enhancement [12,85,86]. Interestingly, these changes have been observed in patients without pre-existing cardiovascular disease, cardiovascular symptoms, or abnormal biomarkers during COVID-19 infection [86]. However, the lack of CMR imaging performed prior to infection means that it is unclear whether the reported abnormalities are actually caused by COVID-19.

### Predictive Factors for Post-COVID Syndrome

The risk factors for post-COVID syndrome and its cardiac manifestations are poorly understood. Research has demonstrated an association between the severity of disease in the acute phase of infection and the likelihood of long-term cardiovascular complications [8]. Patients who are treated in intensive care have a greater risk of cardiac disorders in the post-acute period than those who were not hospitalised [8], likely due to the lasting effects of severe acute cardiac damage. However, the same study showed that the risk was also present in patients who were not hospitalised [8], as supported in the wider literature [12]. Asymptomatic COVID-positive patients may also develop post-COVID syndrome and its cardiac symptoms [87]; thus, the severity of acute infection alone is not a reliable predictor of heart complications in the post-acute phase.

There appears to be an association between female sex and persisting COVID-19 symptoms. A global meta-analysis indicated that men are significantly more likely to necessitate ITU admission or die from COVID-19 [88], but several studies suggest that those of female sex are at greater risk of developing post-COVID syndrome [78,89,90,91]. The COVID Symptom Study application data found that participants experiencing symptoms at 4 to 8 weeks were more likely to be women [78]. Indeed, female sex has been identified as a predictor of worse patient-perceived recovery [91], with self-reported rates of recovery at 3 months after discharge in patients under 50 years being five times lower in women than in men [89]. Despite this, it is unclear whether women are more likely to experience the cardiac symptoms of post-COVID syndrome. Of interest, the female predisposition for long-term symptoms following an infectious disease has also been noted in Lyme disease [92], although the mechanisms underlying this are unclear. Further work is required to delineate sex-related differences in post-COVID syndromes, including long COVID.

One of the proposed causes of long COVID is a failure to rapidly respond to the initial viral infection due to deficiency in the immune system. Low IgG3 levels during initial COVID-19 infection has been associated with an increased risk of developing long COVID [93]. Patients with long COVID were also observed to have lower IgM levels during acute infection, as well as after 6 and 12 months [93]. This highlights the importance of immunoglobulins in managing long-term prognosis from COVID-19 and has the potential to guide identification of future therapeutic targets for post-COVID syndromes.

## 5. Conclusions

The pandemic has shown that COVID-19 is a multi-system disease. Whilst it is unsurprising that a severe inflammatory disease can often trigger cardiac complications in patients with pre-existing cardiovascular risk factors, reports of younger and less vulnerable individuals experiencing acute and chronic heart abnormalities are both unexpected and concerning. The pathophysiology of COVID-19 is still being extensively researched, and new findings may modify the existing classification of low- and high-risk COVID-19 patients.

Whilst the incidence of cardiac complications in non-comorbid patients may be considered low, its significance should not be underestimated when considering the potential impact on healthcare systems. Clinicians should be aware of the unexpected cardiac symptoms reported by low-risk patients and consider the possibility of an underlying COVID-19 infection. With COVID-19 still common throughout the world, and the potential for further waves and new variants in the future, the burden of COVID-19 in cardiology remains substantial. The management and follow-up of post-COVID-syndrome patients is likely to contribute to the ever-increasing demand for general practitioners and primary care services.

The evidence discussed in this review highlights that the pathophysiology of COVID-19-associated cardiac complications is not completely understood. By gaining a greater understanding of the mechanisms underlying the effects of COVID-19, scientists may be able to identify new predictors of poor outcomes and develop treatments for the cardiac consequences of COVID-19.

## Figures and Tables

**Figure 1 viruses-14-01322-f001:**
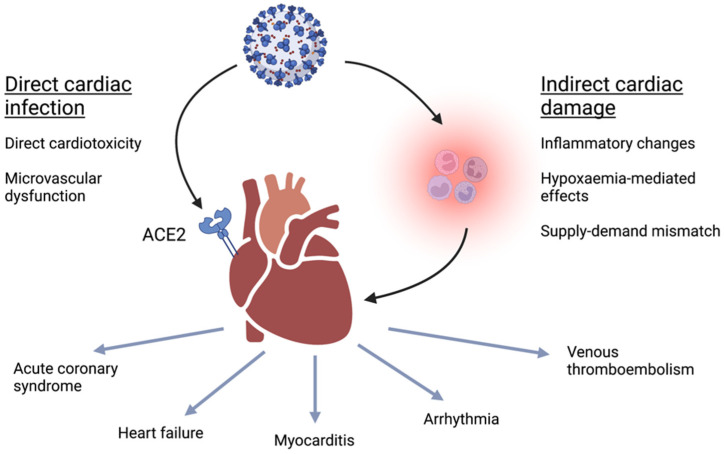
An illustration of the potential mechanisms underlying COVID-19 complications in the heart. A schematic of direct and indirect mechanisms through which COVID-19 causes cardiac complications. The COVID-19 virus (shown at the top of the figure) may directly bind to cardiomyocytes and endothelial cells (left) to cause direct cardiac infection and cardiotoxicity. Indirect damage may also occur (right), with downstream inflammatory and hypoxaemic mechanisms. In the lower figure, we show the consequent cardiac complications that may result. ACE2 = angiotensin-converting enzyme 2. Created with BioRender.com with permission.

**Table 1 viruses-14-01322-t001:** A summary of the main differences between paediatric inflammatory multisystem syndrome temporally associated with COVID-19 (PIMS) and Kawasaki disease.

	PIMS	Kawasaki disease
**Age**	School-age children (~7–8 years)	Infants and young children (~3 years)
**Ethnicity**	Predominantly Black and Hispanic	Asian
**Gastrointestinal involvement**	Common	Uncommon
**Cardiovascular involvement**	Common	Less common

PIMS = paediatric inflammatory multisystem syndrome temporally associated with COVID-19.

## Data Availability

Not applicable.

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
