# Peer review of "Cardiac Complications of COVID-19 in Low-Risk Patients"

_viruses, 2022, doi:10.3390/v14061322_

Round 1
Reviewer 1 Report
I enjoyed reading this well written and timely review.
Some suggestions for clarity:
Line 207. "Since initially asymptomatic patients.." Instead "asymptomatic COVID positive patients..."
Table 1. Define PIMS in Table.
I noticed some grammatical errors in lines 94, 117, 298-299.
Author Response
Line 207. "Since initially asymptomatic patients.." Instead "asymptomatic COVID positive patients..."
Response: Many thanks. We have amended in new line 303.
Table 1. Define PIMS in Table.
Response: We have added this in full in the table title (line 199) and also in the subtext to the table (line 202).
I noticed some grammatical errors in lines 94, 117, 298-299.
Response: Many thanks for identifying these. They are now corrected.
Reviewer 2 Report
The review fits well a new field of interest in cardiology which is evaluation of COVID-19 complications and requires careful consideration.
Major comments:
1.The structure of review should be changed as it should start with pathophysiological aspects and underlying mechanisms leading to cardiac complications. Figure 1 is too simply, moreover does not present other important mechanisms like direct cardiotoxity, hypoxemia-mediated effects, supply-demand mismatch etc. I suggest to improve this figure by adding other mechanisms and adding cardiovasular manifestations caused by these effects i.e. ACS, arrythmia, myocarditis, VTE and HF.
2. Authors should distinguish and describe thoroughly at least three fields of interest for cardiologists with regard to SARS-CoV-2 infection:
— acute cardiac manifestations of SARS-CoV-2 infection (acute coronary syndrome, exacerbation of heart failure, myocarditis, cardiac arrhythmias and their exacerbation),
-post-COVID syndromes, most often: abnormal heart rhythm, signs of myocarditis, vasculitis); and finally
- chronic organ (cardiac) damage associated with LONG COVID syndrome.
3. There are no diagnostic approaches discribed (MRI, cardiac biomarkers, inflammatory biomarkers in acute and chronic state)
5. What is the conclusion for physicians, cardiologists, internists and general practitioners?
Author Response
1.The structure of review should be changed as it should start with pathophysiological aspects and underlying mechanisms leading to cardiac complications. Figure 1 is too simply, moreover does not present other important mechanisms like direct cardiotoxity, hypoxemia-mediated effects, supply-demand mismatch etc. I suggest to improve this figure by adding other mechanisms and adding cardiovasular manifestations caused by these effects i.e. ACS, arrythmia, myocarditis, VTE and HF.
Response: Many thanks for your comments. We have rearranged the manuscript as you suggest. We have altered Figure 1 and adjusted the text accordingly. Figure 1 now shows these alternate mechanisms and downstream complications that may occur as discussed. Further, we have added a full explanation into the Figure legend.
- Authors should distinguish and describe thoroughly at least three fields of interest for cardiologists with regard to SARS-CoV-2 infection:
— acute cardiac manifestations of SARS-CoV-2 infection (acute coronary syndrome, exacerbation of heart failure, myocarditis, cardiac arrhythmias and their exacerbation),
-post-COVID syndromes, most often: abnormal heart rhythm, signs of myocarditis, vasculitis); and finally
- chronic organ (cardiac) damage associated with LONG COVID syndrome.
Response: Many thanks. We edited the manuscript to discuss the fields in greater detail. Notably, we discuss:
- Acute coronary syndrome on lines 80-82, 129-130 and 166-168
- Heart failure on lines 160-163 and 248-256
- Arrhythmias on lines 49-50, 91-94, 111-113, 129-130 and 183-184
- Arrhythmias in long COVID on lines 273-281
- Myocarditis and ongoing inflammation in long COVID on lines 253-256, 70-272 and 284-286.
- There are no diagnostic approaches discribed (MRI, cardiac biomarkers, inflammatory biomarkers in acute and chronic state)
Response: We agree with the reviewer and have added paragraphs about diagnostic approaches to the acute and post-acute complications sections. These paragraphs for MRI, cardiac biomarkers and inflammatory biomarkers in acute and chronic state are lines 156-168, lines 214-221 and lines 282-292.
- What is the conclusion for physicians, cardiologists, internists and general practitioners?
Response: We have adjusted the conclusion to directly address clinicians.
Round 2
Reviewer 2 Report
Authors substantially improved their review which satisfy my expectations. I do not have more comments.